# Spatial–Temporal Dynamics of Forest Extent Change in Southwest China in the Recent 20 Years

**Yanlin Zhang** [1], **Shujing Wang** [2,3] **and Xujun Han** [2,3,*]

1 School of Earth Sciences and Spatial Information Engineering, Hunan University of Science and Technology, Xiangtan 411201, China; zhangyanl02@163.com
2 Yellow River Institute of Eco-Environmental Research, Yellow River Basin Ecology and Environment Administration, Zhengzhou 450004, China
3 Chongqing Engineering Research Center for Remote Sensing Big Data Application, School of Geographical Sciences, Southwest University, Chongqing 400715, China
* Correspondence: hanxujun@swu.edu.cn

**Abstract:** Deforestation is thought of as a huge threat to carbon neutrality and the development of contemporary society and it has brought wide interest and attention in the science community to develop new methods to identify and quantify the occurrence and extent of forest loss. Understanding the forest-loss patterns is essential for forest management and protection. With the help of a high-spatial-resolution remote-sensing dataset on forest loss, the spatial and temporal dynamics of deforestation patterns in forests of Southwest China (SWC) have been investigated. The major findings of this study indicated that small-scale (<5 ha) deforestation raised pervasively in the region from 2001 to 2019, and the number of large patches of forest loss (>5 ha) has decreased significantly during the same period. Moreover, the mean size of forest-loss patches showed an increase from 0.34 ha to 0.61 ha over time. With the alarming trend of increasing deforestation in the southern region of our study area, the growth of emerging forest-loss hotspots was clearly observed in Chongqing and Sichuan Province. The results promoted an indepth understanding of forest-loss patterns in SWC and can help provide more coherent guidance for further forest monitoring and conservation.

**Keywords:** deforestation; emerging hotspot; Southwest China

## 1. Introduction

Deforestation is thought of as a major issue and a huge threat to worldwide carbon neutrality. In the last decades, tremendous changes have been observed and recognized in land use or land cover worldwide [1,2], including a significant reduction of forest area. Generally, the reduction of forests is mainly caused by the expansion of agricultural land (both commercial and subsistence) [3], population growth [4], timber demand [5], and mining development [6,7]. Deforestation has multiple environmental consequences [8,9], especially critical impacts on climate. It affects the global environmental ecosystem by increasing a region's carbon emissions [10–13], resulting in biodiversity loss [14–16], and reducing its ability to provide ecosystem services [17]. Therefore, the main goal of national and local governments to improve land management policies is to suppress deforestation and promote forest restoration as much as possible.

The technology and datasets of remote sensing have been widely used in forest inventory, change detection, and management, as they can provide timely and accurate forest-dynamic information [18–23]. Many forest-monitoring systems have been developed globally and regionally based on earth observation satellite images, e.g., the Global Forest Watch platform (http://www.globalforestwatch.org/ (accessed on 10 June 2023)), which provides spatially detailed annual estimates of deforestation information with global consistency, and the PRODES program (Monitoramento do Desmatamento na Amazônia Legal por Satélite). All these kinds of products derived from satellite, airplane, or UAV

images have fundamentally changed the way of monitoring forest dynamics globally. The high spatial resolution (30 m) GFC (global forest change) product developed and published by Hansen et al., provides the possibility for large-scale and accurate assessment of forest-loss dynamics (both spatial and temporal) [24]. Most studies that have used GFC or other forest survey data focused on the area of deforestation or its influencing factors and omitted the spatial structure and pattern [3,25–28]. In the context of forest protection, the complexity of deforestation requires new analysis methods. Although spatial pattern analysis is not inherently predictive, it can help identify the spatial and temporal trends of forest change, so as to quickly and consistently determine the priority of management intervention [25,29]. It has been used in many fields, such as determining fire hotspots [30], evaluating population aging trends [31], and species biodiversity assessment [32].

Dominated by mountainous areas, Southwest China (SWC) has a very high spatial coverage of forest. With the global climate warming and rapid economic development of China, the land use and ecosystem in SWC have had a dramatic change, which makes it the case study area. Although much research on forest-cover change in SWC has been carried out in the last decades, the pattern and process of deforestation in the 21st century in this region still remain unclear. Some interesting questions still need to be answered, e.g., have there been changes in patterns of deforestation over time? Have there been changes in the geometrical morphology of forest-loss area or patches? Can the hotspots of forest change be quickly identified? Considering these above problems and improving our understanding of the dynamic changes of forests in SWC, this study makes use of landscape metrics and emerging hotspot analysis to comprehensively identify the spatial and temporal patterns of forest loss in SWC during the first two decades in the 21st century (i.e., from 2001 to 2019) based on the GFC product. The objectives of our work were focused on: (1) the size of forest-loss patches, (2) the landscape metrics trend of forest loss, and (3) emerging forest-loss hotspots. This study will help to deliver indepth insight into the process of deforestation in SWC and assist in promoting effective measures for forest conservation going forward.

## 2. Materials and Methods

### 2.1. Study Area

In this study, SWC is defined as an area consisting of four provinces (Chongqing, Sichuan, Yunnan, Guizhou, and Guangxi) in Southwest China (20°54′~34°19′ N and 97°31′~112°04′ E). Denominated as a mountainous area, it covers over 136,400,000 ha of land area, with rich forest resources and biodiversity. The distribution of forests in 2000 is shown in Figure 1. It is also a typically karst-intensive region, as it has the largest karst area in the world (54,000,000 ha). Restricted by its geological background, the landforms in this region are susceptible and sensitive to land cover change and the ecosystem environment is extremely fragile. In the late 20th century, promoting the planting of fast-growing trees (mainly eucalyptus and poplar) in the area not only increased vegetation coverage but also stimulated the development of related economic industries. According to the statistical yearbook, in 2018, forests accounted for 47.6% of the land area in SWC. By the end of 2019, the population reached 249.4 million, and the gross domestic product (GDP) grew up to 13.145182 trillion China Yuan, with the per capita GDP increasing 10.8 times compared with that in 2000. Such a fast development in the economy and complex geographic characteristics will inevitably exert remarkable influences on land use patterns. In recent years, the fragile karst ecological environment has made the problem of rocky desertification in this area more and more prominent. Local farmers destroyed forests to grow food, leading to serious vegetation degradation and soil erosion; therefore, the forest ecosystem has suffered serve disturbances.

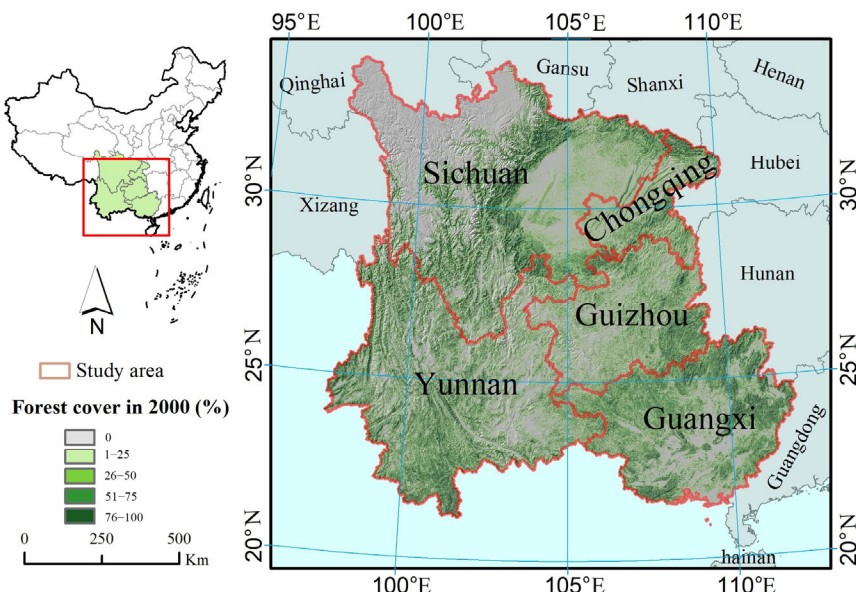

**Figure 1.** Location and tree cover of Southwest China.

## 2.2. Data and Methods

### 2.2.1. Global Forest Change Data

GFC is a dataset developed to monitor the global forest dynamics in the twenty-first century, based on over 600 thousand Landsat 7 and Landsat 8 images in growing seasons [24]. The dataset includes the percentage coverage of trees in 2000, the annual forest loss per pixel from 2001 to 2019, and the forest gain from 2001 to 2012. In this dataset, forest loss is defined as a complete removal of tree canopy (i.e., the tree cover decreased to 0% or a specified threshold), usually caused by logging and fire, and leading to conversion of land-cover type. In the dataset, a pixel of 30 m × 30 m is either encoded as 0 (standing for no forest loss), or the year when the forest in the pixel is completely gone (e.g., the pixel value is specified between 2001 and 2019). Forest gain is encoded as a binary value in each pixel, indicating whether forest gain occurred during the period from 2000 to 2012. The GFC dataset had already been validated by FAO statistics, and other LiDAR and satellite-inferred products, showing an overall accuracy higher than 90%. In this study, we used the GFC dataset (Version 1.7) to asses changes in forests from 2001 to 2019, available online at Google Earth Engine (http://earthengine.google.com/ (accessed on 10 June 2023)), which is a powerful platform that enables geospatial data high-speed analysis, using advanced processing tools for large regions and large datasets, without having to download the remote sensing data for processing The analysis boundaries were restricted to Southwest China, generating a mosaic by using the spatial limits of the SWC, and then the images were converted from the WGS84 coordinate system to a projected coordinate system to obtain the area measurements for forest-loss patches. The tree coverage in 2000 was chosen as a starting point of the analysis, and a threshold of 20% for tree coverage was used to generate a binary forest–nonforest mask according to pixel values [33]. In this study, a yearly series of forest losses were extracted from the GFC dataset during the period from 2001 to 2019 by using this mask.

### 2.2.2. Analysis of Landscape Metrics for Dynamics of Forest-Loss Patches

To investigate the spatiotemporal pattern of forest loss in geometric attributes over time, a fishnet grid with 14,196 cells (10 km × 10 km) was created over the study area [34]. We used spatial-landscape metrics to capture the spatial and temporal patterns of change in forest loss throughout the SWC by focusing on the deforestation process. Four landscape metrics were chosen and used in each grid cell, including the total area of forest loss, counts of forest-loss patches, the mean patch size (i.e., the mean area of patches in a cell), and edge

density (calculated as the edge length per unit area) [35]. Patches represent areas where forest-loss events occur in our study. Forest-loss patches were classified into eight categories: with area less than 1 ha; area $\geq$ 1 and <5 ha; area $\geq$ 5 ha and <10 ha; area $\geq$ 10 ha and <50 ha; area $\geq$ 50 ha and <100 ha; area $\geq$ 100 ha and <200 ha; area $\geq$ 200 ha and $\leq$500 ha; area greater than 500 ha. This categorization criterion was proposed by Rosa, et al., and slightly modified, taking into account the actual situation in SWC [36]. The changing trend (increase or decrease) of landscape metrics for each grid cell was tested with a nonparametric Mann–Kendall (MK) method [37], in which the samples are unnecessary to fit a specific distribution. In this work, significant trends are defined as those at the $\alpha$ = 0.01 signification level, and the MK test was calculated as follows:

$$S = \sum_{i=0}^{n} \sum_{j=i+1}^{n} sign\left(y_j - y_i\right) \tag{1}$$

where $n$ stands for the sample size (of observations); $y_i$ or $y_j$ are observed values in given years (note that here $y_i$ is earlier than $y_j$). Generally, a large positive value of S means an increasing trend and a large negative value means a decreasing trend. A small absolute value of $S$ indicates that the trend is insignificant, or absent. The absence of trend was set as the null hypothesis, and it was tested with Z values calculated using an R package.

To investigate and visualize the changes in the spatial density for forest-loss patches, the percentage of deforestation for each grid cell was calculated on the basis of forest coverage in 2000. Each grid cell was classified into one of five categories following Kalamandeen et al. [38], i.e., negligible (with the percentage of deforestation <0.01%), light (between 0.01% and 0.1%), moderate (0.1%–1%), heavy (1%–10%), and very heavy (>10%).

### 2.2.3. Emerging Hotspot Analysis (EHA)

EHA is aiming at identifying statistically significant trends in a dataset and obtaining the spatial positions with high-value or low-value elements clustering [29]. The EHS is usually conducted using two statistical methods jointly to evaluate the spatial–temporal patterns. The first is the Getis–Ord Gi* statistic, which is used to determine the location of spatial clustering as well as its degree [39]. The other is Mann–Kendall trend analysis, which is used to evaluate the temporal trend of the data series over time [37]. In this study, hotspots are defined as areas (or pixels) with statistically significant spatial clustering for deforestation. The EHA was conducted with the help of the Emerging Hotspot Analysis tool in Arcgis 10.7.

First, before the analysis, the dataset is transformed into a space–time cube (in the format of NetCDF, i.e., network common data form). Then, the forest-loss points or pixels in each province were aggregated into space–time bins, with a spatial resolution of 10 km. In EHA, Getis–Ord Gi* runs at a fixed distance (10 km). The features located within this distance are recognized as its neighbors and those further away are not. The counts of mapped deforested cells (30 m) in each bin were used to calculate the Getis–Ord Gi* statistic as follows, i.e., Equation (2),

$$G_i^* = \frac{\sum_{j=1}^{n} \omega_{i,j} \cdot x_j}{\sum_{j=1}^{n} x_j} \tag{2}$$

where $x_j$ is the number of deforested cells (30 m) within bin $j$. $\omega_{i,j}$ is a weight matrix. In the matrix, the element values are equal to 1 for bins in a distance less than 10 km from bin $j$. Otherwise, the value is set to 0. $n$ is the sample size for mapped deforestation bins.

A Z-score and the corresponding $p$ value were calculated to assess whether the forest loss in some specific neighboring bins is statistically significant compared to other bins in the study region. The sign of the calculated Z score can tell that the trend in a bin is increasing (with a positive Z score) or decreasing (with a negative Z score). The combinations of a large positive Z score (e.g., $\geq$1.96) with significant $p$ values (e.g., <0.05) generally mean local clusters of a large area and severe forest loss (hotspot). Similarly, combinations of low negative Z scores (e.g., $\leq$−1.96) with significant $p$ values (<0.05) mean local clusters of small forest-loss areas (coldspot).

Second, the MK test was applied to assess whether there exists a significant trend in the series of Z-scores calculated from Equation (2) for each bin. The calculated results from Equation (2) and MK trend analysis were then jointly used to classify each bin into one of 17 categories (by default), one nonsignificant category, eight hotspot categories, and eight coldspot categories. Every category represents a different combination of spatial clustering type and temporal trend. However, redundant information in the output maps may distract attention from the relevant results. Here, hotspot trends were focused on rather than displaying all the results (nonsignificant trends and coldspots were not displayed). Definitions for four hotspot categories are listed in Table 1.

**Table 1.** Interpretations for four hotspot categories in the Emerging Hotspot Analysis.

| Hotspot Category | Definition |
| --- | --- |
| New | Locations where are significant hotspots statistically at the final time step, and they have never been significant hotspots at other time steps. |
| Sporadic | Locations appear as on-and-off-again or off-and-on-again hotspots. Less than 90% of the time steps should be significant hotspots, and there are no significant coldspots at any time steps. |
| Consecutive | A location where there is only one single (uninterrupted) run of significant hotspots in the last consecutive time steps. Besides, there should be no significant hotspots prior to the appearance of the final hotspot run, and less than 90% of the time steps are significant hotspots. |
| Persistent | A location where there has been a significant hotspot persistently, or in more than 90% of the time steps, and there is no discernible increase or decrease in the intensity of clustering over the analyzing period. |
| Oscillating | A location where there is a significant hotspot at the final time step, and there also exists a significant coldspot at a prior time step. Of course, significant hotspots occur in less than 90% of the time steps. |

## 3. Results

### 3.1. Spatial and Temporal Patterns of Forest Loss during 2001–2019

According to the dataset, the total area of forest loss in SWC is about 3.7527 million ha from 2001 to 2019. However, the area of forest loss is different over the years (as shown in Figure 2). During the period from 2001 to 2008, the area of forest loss showed a rapid increase. After reaching a peak in 2008, the forest loss fluctuated and decreased until 2015, and then increased greatly in 2016 and 2017, remaining at a high level of deforestation. The mean patch size of forest loss also seems interesting. The average area of forest-loss patches showed a similar time pattern to the total forest-loss area in the region. During the period from 2001 to 2005, with a small deforestation area, the mean patch size is also small, less than 0.5 ha. The mean patch size increased by 83.69% from 2001 to 2008. After that, with some fluctuations, the growth rate of patch size does not further accelerate between 2014 and 2016. From 2014 to 2016, the mean patch size increased by 65.29%. In 2016, the mean patch size reached the maximum value of 0.73 ha. At the same time, the total forest-loss area reached a peak value of about 297,311 ha. Overall, the mean forest-loss patch size increased across Southwest China from 0.34 ha to 0.61 ha from 2001 to 2019 but varied considerably across provinces. In terms of forest gain, the previous GFC datasets update was in 2012, which might bias the actual growth of forests in the region. From 2001 to 2012, the forest area increased by a total of 864,300 ha and the main forest-gain patch size is 2.1 ha.

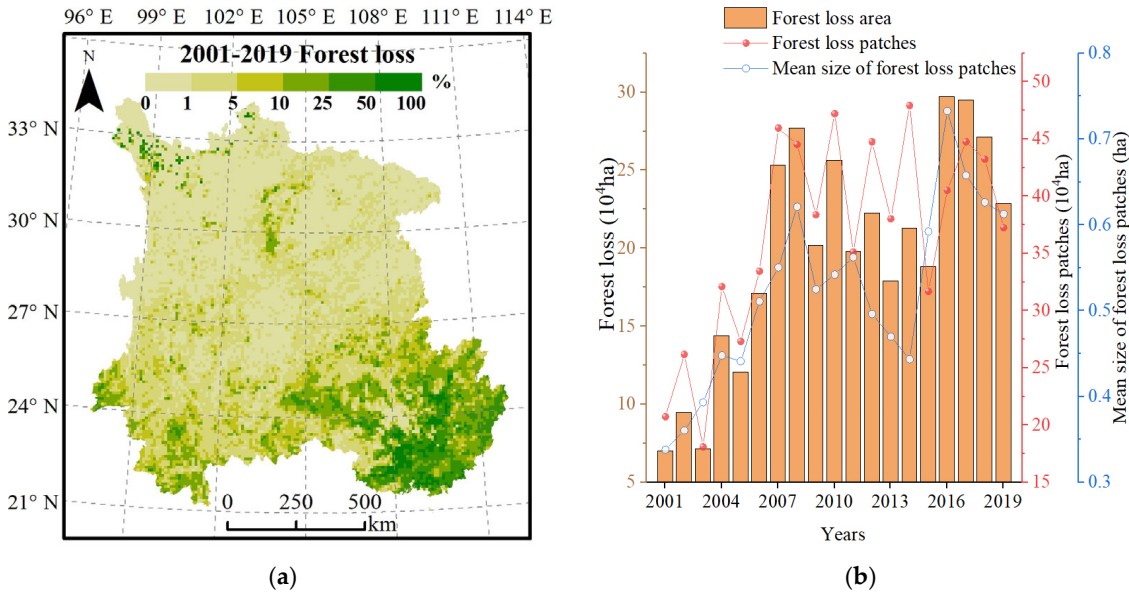

(a)

(b)

**Figure 2.** Spatial–temporal evolution of forest loss in Southwest China (SWC) from 2001 to 2019. (**a**) the proportion of forest cover affected by forest loss in each 10 km × 10 km cell; (**b**) annual averaged forest loss, with the number of patches and their mean patch size illustrated.

The density of forest loss has been calculated at each 10 km × 10 km cell in the fishnet grid, describing the total deforestation area in each grid cell during some specified periods in the 19 years (from 2001 to 2019). It is found that the geographical patterns of deforestation density were different over time (Figure 3). From 2001 to 2007, 12.7% of cells in the study area fell into the negligible forest-loss category (<0.01%). Cells falling into categories of light (0.01%–0.1%) and moderate (0.1%–1%) forest loss accounted for about 26.5% and 35%, respectively. A decreasing trend was observed for the total number of grid cells that experience negligible, light, or moderate deforestation, from 74.3% in 2001–2007 to 68.4% in 2008–2013. In contrast, the number of cells classified as heavy (1%–10%) and very heavy (>10%) deforestation took an increasing trend. The proportion of cells for these two categories is about 16.8% in 2001–2007, 22.9% in 2008–2013, and 21.4% in 2014–2019, respectively.

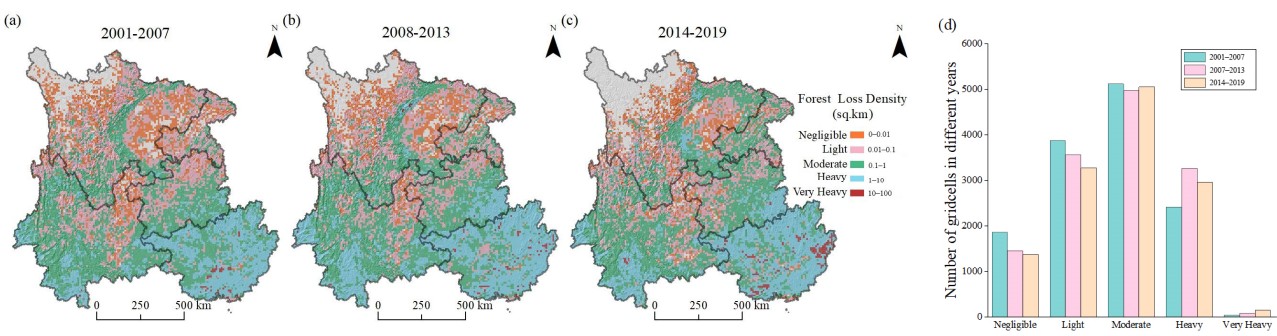

**Figure 3.** The density of forest loss (%) in Southwest China based on the GFC product for three periods. (**a**) from 2001 to 2007; (**b**) from 2008 to 2013; (**c**) from 2014 to 2019 and (**d**) the number of cells for each category of forest-loss density, shown in the histogram.

The number of forest-loss patches with an area of less than 5.0 ha accounted for about 98.8% of all patches and the area for the majority of patches (about 90.5%) was less than 1.0 ha (Figure 4). The number of patches with an area less than 1.0 ha increased significantly during the study period, especially reaching a peak value in 2014. The total area of patches with the area less than 1.0 ha and 5.0 ha accounted for about 41.7% and 32.0% of total forest loss, respectively. From 2008 to 2015, the number of forest-loss patches of various

size scales showed a decreasing trend, because the total forest loss decreased (Figure 2b). The decrease was mainly (48.5%) caused by the reduction in forest-loss patches with an area larger than 5.0 ha. After 2016, the number of deforestation patches with different sizes also showed a decreasing trend. The forest loss in SWC exhibits a pattern of small-scale fragmentation, which indicates that the forest loss in SWC is not mainly caused by mechanized activities from the outside. Instead, it is mainly driven by local people from the surrounding nonforest areas.

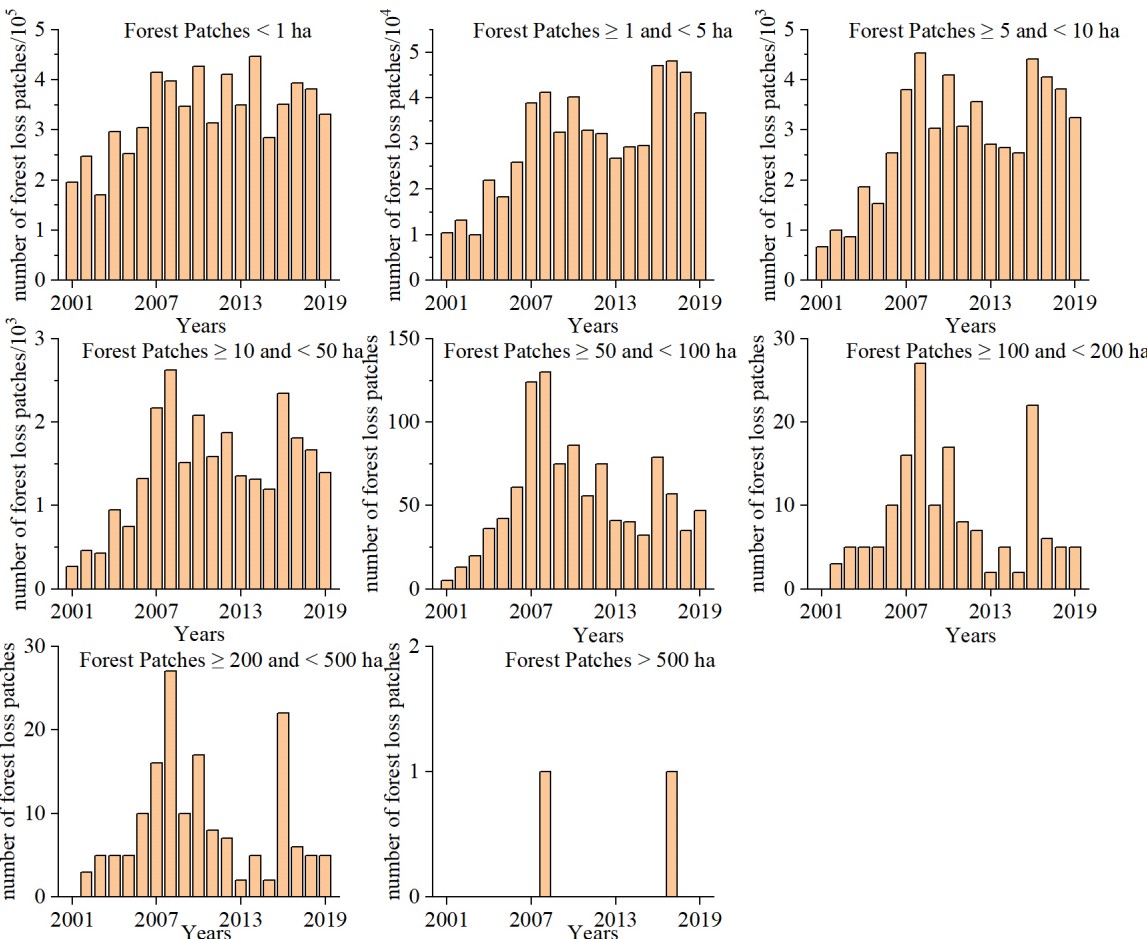

**Figure 4.** Distribution of forest-loss patch size in Southwest China from 2001 to 2019.

According to the analysis results of the 19 years, the grid-level forest-loss occurrences in SWC from 2001 to 2019 detected severe deforestation in the southern of SWC. In addition, low-density and small-scale deforestation is gradually eroding Southwest China, especially Chongqing and Sichuan Province, despite overall augments of large-scale patches of deforestation in Guangxi Province.

### 3.2. Temporal Trend of Landscape Metrics of Forest Loss

According to the spatially distributed trends of significant forest loss, the forest loss showed an increasing trend in the marginal areas of the Sichuan Basin, the surrounding areas of Guangxi Province, and the tropical rain forest in Yunnan Province (Figure 5a). In total, 28.27% of the cells showed a rising trend for the forest-loss area, and only 9.43% of the cells showed a decreasing trend in the forest-loss area, mainly located in Western Sichuan Province; 25.17% of the cells showed that the patch number of forest loss was increasing (Figure 5b), and the fraction of cells with a decreasing trend for the patch number of forest loss was close to half (12.56%). In terms of the mean patch size, an increasing trend was found in 31.38% of cells, and this is a combined result of increased forest-loss area and the

number of patches (Figure 5c). The spatial distribution of these cells has a good consistency with the areas of increasing forest loss. In Southern Guangxi Province, the mean patch size decreased significantly from 2001 until 2019. Overall, the patch size in about 7.85% of the fishnet cells was decreasing. However, the trend for edge density showed different results. The study indicated that the edge density in 24.95% of cells declined significantly over time (Figure 5d), which is driven by the increase of patch number of forest loss and deforestation area. The edge density of forest-loss patches with more patch numbers is smaller than that with fewer patches. Only 4.97% of grid cells have an increasing edge density.

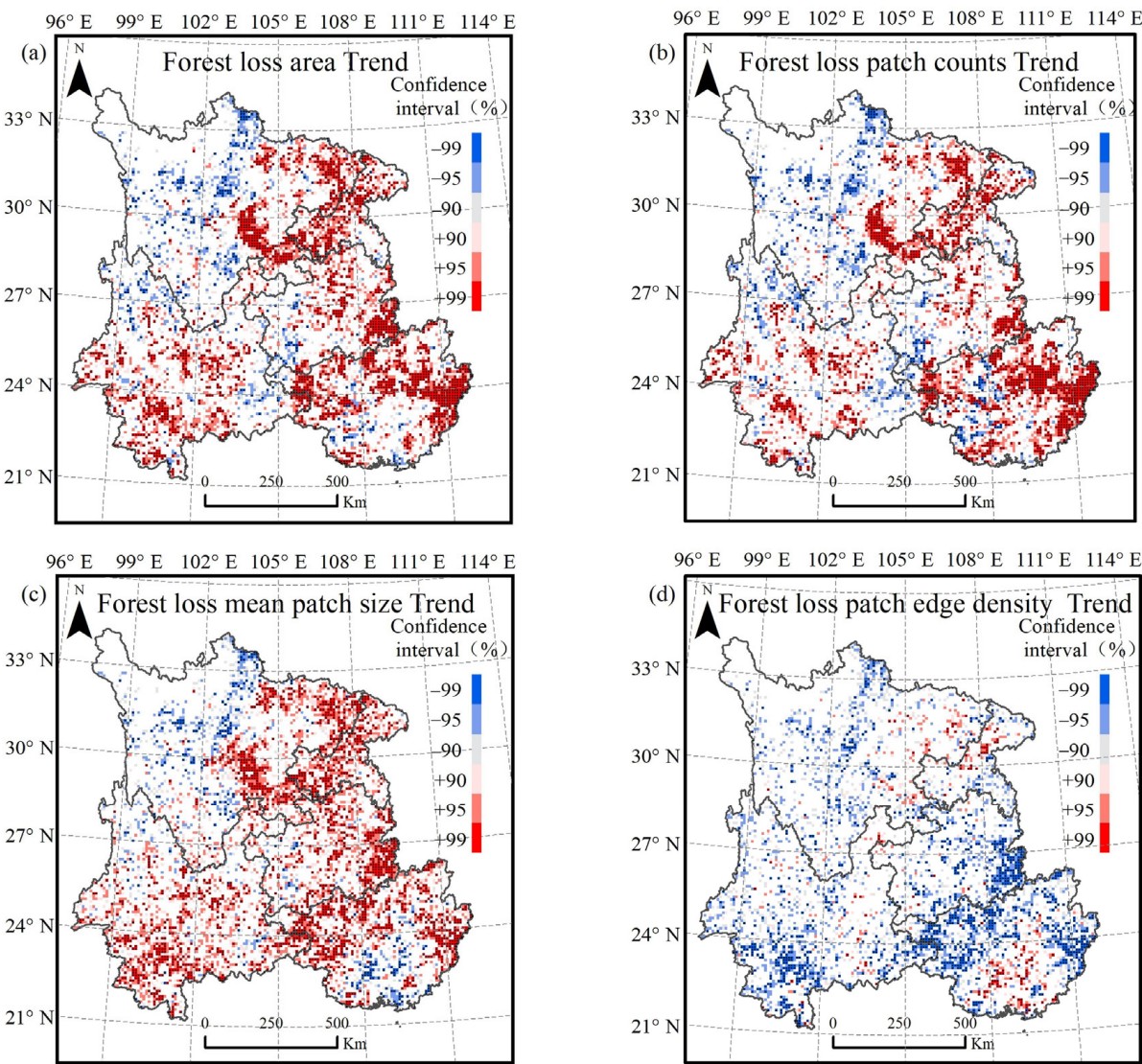

**Figure 5.** Temporal trend of landscape metrics of forest loss in Southwest China (SWC). (**a**) Trend for forest-loss area in each grid; (**b**) Trend for the number of patches with forest loss in each grid; (**c**) Trend for the mean patch size in each grid; (**d**) Trend for the edge density in each grid. The analyzed time series is from 2001 to 2019 and the confidence levels for the trend were represented by using Z values.

### 3.3. Evolution of Forest-Loss Hotspots

The emerging hotspot analysis indicated that, from 2001 to 2019, deformation clusters were more concentrated in the northeastern and southern parts of the SWC (Figure 6). A great number of grid cells experiencing deforestation were sporadic in nature (63.16%), which is followed by the number of cells for consecutive hotspots (30.37%), new hotspots (5.66%), oscillating hotspots (0.67%), and only two persistent hotspots. New hotspots with statistically significant deforestation shifted gradually toward the northeastern region of

the study area, especially in Sichuan Province and Chongqing, which means that there were more forest-loss patches aggregated in recent years. At a province level, 27.19% of the grid cells in Guangxi Province were represented by forest-loss hotspots, followed by 15.61% in Chongqing and 7.62% in Guizhou Province. The proportion of hotspot cells for Yunnan and Sichuan Province is 6.86% and 4.95%, respectively. The distribution of various hotspots in each province is shown in Table 2.

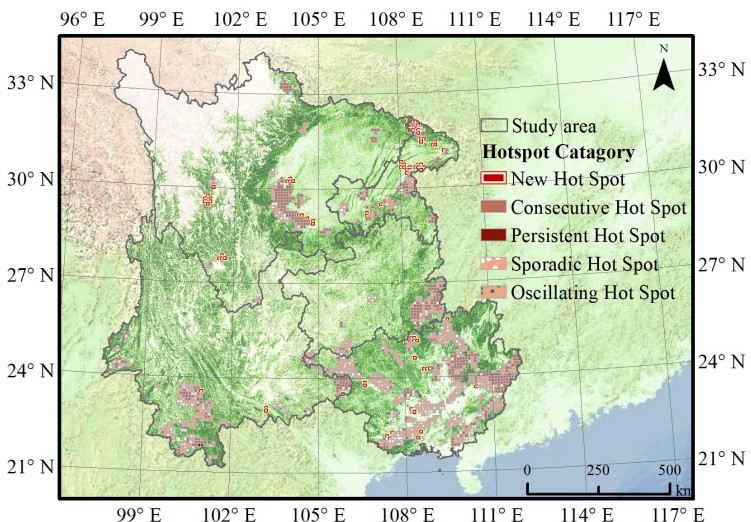

**Figure 6.** Hotspots map of forest loss identified from the spatiotemporal trends in GFC datasets.

**Table 2.** Province-wise proportion distribution of emerging hotspots in Southwest China.

| Province | New Hotspot | Sporadic Hotspot | Consecutive Hotspot | Persistent Hotspot | Oscillating Hotspot |
|---|---|---|---|---|---|
| Chongqing | 37<br>25.17% | 62<br>42.18% | 48<br>32.65% | | |
| Sichuan | 20<br>9.35% | 81<br>37.85% | 113<br>52.80% | | |
| Yunnan | 4<br>1.42% | 199<br>70.57% | 77<br>27.30% | 2<br>0.71% | |
| Guizhou | | 89<br>60.54% | 58<br>39.46% | | |
| Guangxi | 23<br>3.31% | 507<br>72.95% | 155<br>22.30% | | 10<br>1.44% |

## 4. Discussion

The way forests are destroyed generally gives information to help intervene in forest exploitation and management. This study investigated the dynamics of forest loss and its spatial patterns by use of landscape metrics and EHA in SWC and helps to predict the evolution of pressures on forest-resources management.

### 4.1. Deforestation Characteristics in SWC

The results of this study report a more complicated phenomenon for deforestation dynamics in SWC than those that have been reported. The results agree with the analysis of Wu et al. [40], e.g., the general increase of forest loss from 2001 to 2008, and then followed by some decrease fluctuations until 2015. As has been reported in previous studies, inhibition of forest loss mainly depends on regional and national policies [41–43]. Especially in 2008, the Chinese State Council approved the "Planning Outline for Comprehensive Treatment of Rocky Desertification in Karst Areas (2006–2015)", partly contributing to the decrease. In a view of the distribution of deforested patch size, the temporal pattern of mean patch size of

forest loss is similar to the total forest-loss area; that is, an increasing number of small-scale patches (<5 ha) was observed across almost all SWC provinces. Small-scale deforestation events were the main performance of forest loss in SWC and showed an upward trend over time, while the larger-scale (>5 ha) forest-loss patches obviously decreased after 2008. But the mean forest-loss patch size overall became larger. Interestingly, Bruno Montibeller et al., found a different result in the Brazilian Legal Amazon [37]. In order to avoid being detected, the patches of forest loss made by local landholders there not only became smaller but also spread widely. However, the mean deforested area of the patches in SWC was much lower than that in the Brazilian Legal Amazon.

The trend analysis for forest density, as well as landscape metrics in cells of a fishnet grid, also revealed the forest-loss dynamics in SWC. A current increasing trend for the deforested area and mean patch size was observed across all the provinces in SWC, typically in Guangxi and Sichuan. The forest-loss density also increased by an order of magnitude or more between 2001 and 2019. It is challenging to attribute reasons for the observed increase in forest loss specifically [44–47]. However, according to some powerful reasons, it is believed that the observed increase in forest-loss events is related to human activities. It should be noted that any forest disturbance in the GFC product is assumed to be forest loss, whether it is selective logging or affected by fire [29]. In this case, the harvest in plantations could also be considered as forest loss in Guangxi, where there are many fast-growing forests.

*4.2. Emerging Hotspots in Southwest China*

Significant forest-loss hotspots in SWC are also examples of deforestation pattern change. The results of this study emphasized the emergence of new forest-loss hotspots in Chongqing and Sichuan Province. In the period from 2001 to 2019, 25.17% of hotspots were defined as new hotspot in Chongqing, primarily in Chengkou regions. This appears to be associated with mining activities, where it is known as the "Western Mining Captial". On 12 November 2013, the Ministry of Land and Resources issued the Announcement on the Establishment of the Third Batch of Integrated Exploration Zones (No. 18 of 2013), which officially listed the "Chengkou County Manganese Mine Integrated Exploration Zone in Chongqing" as the third batch of integrated exploration zones in China. The Chengkou Party Committee and Government have vigorously developed mining and actively promoted the construction of new industrialization. In 2017, Chongqing Chengkou Industrial Park purchased 80 acres of land and invested 67.8 million China Yuan to launch an annual production of 100,000 tons of barium sulfate project. In Sichuan, new hotspots mainly appear near the Liangshan and Yalong River basins. The recent increase in forest fires can partly explain the forest loss in Liangshan [30]. The dry and hot climate there increases the risk of forest fires, in turn resulting in forest loss. Many studies have shown that there is a strong relationship between them [40,48]. In addition, large-scale infrastructure construction is also a driving factor in promoting deforestation hotspots in Sichuan. The development and construction of hydropower stations along the Yalong River Basin began in 2003 and it is estimated that 21 cascade hydropower stations that are a combination of large and medium-sized with good reservoir regulation performance will be built before 2025. The completion of the high-speed railway on the southwestern edge of the Sichuan Basin from Chengdu to Yibin in 2019 may be associated with the subsequent forest loss in this region as well. From 2001 to 2019, the average rate of forest loss in Guangxi is about 120,010.7 ha yr$^{-1}$, although the overall rate declined in recent years. Forest loss widely spread across the entirety of the region, as evidenced by the sporadic hotspots and consecutive hotspots there. Due to the large areas of plantations, the results of hotspots in this region will become complicated, which may reflect the harvest cycles within plantation boundaries. In addition, the economic development in Xishuangbanna in Yunnan Province has promoted the cultivation of commodity agriculture such as rubber and tea in the last two decades, hence inevitably resulting in significant forest loss [49]. Many of the sporadic hotspots intersected with the widely known tropical rainforest area in these regions potentially suggesting that the primary forest is being destroyed there. Where other hotspots

overlapping with the locations of tree plantations likely reflects that the plantation-harvest cycles of eucalyptus and pine are prevalent in the province.

### 4.3. Socioeconomic Correlates of Forest Change

On a global scale, both historical and recent tree-cover loss followed clear spatial and macroeconomic patterns. Spatially, losses have disproportionately affected areas with high population pressure and easy access. In low-income areas, a large fraction of tree cover still coincides with areas of high population pressure. The proportion of tree-cover losses not compensated by within-country tree-cover gains was highest in countries with high levels of poverty, urbanization, population growth, a high GDP reliance on agriculture, and with expanding food production. China has one of the world's highest correlations between area suitability for tree cover and for agriculture (globally R = 0.35; $p \leq 0.001$; in China R = 0.73, $p \leq 0.001$) and the zone that is suitable for tree cover, therefore, overlaps to 84% with the zone suitable for agriculture. As is the case globally, both historical and recent forest loss in SWC has mostly affected areas characterized by rapid expansion, high population pressure, poverty, and an emphasis on food production. The increase in population will increase the demand for agricultural products, especially in economically backward mountainous areas; it is more likely to cause deforestation. The Pearson correlation coefficient between permanent population and forest loss reached 0.77 ($p < 0.01$), and the gross agricultural product is 0.71 ($p < 0.01$), the GDP of SWC increased by 10 times from 1182.13 billion to 13,119.98 billion China Yuan from 2005 to 2019. Given SWC's population size and rate of economic growth, a realistic understanding of China's forestry situation is important as SWC is responsible for a large part of China's 'real estate' and imports more timber and exports more wood products than any other region. Some key forest programs have achieved some major successes, such as greatly reduced logging in the target provinces of the National Forest Protection Program, Grain for Green programs, and a massively expanded protected forest-area network. The aims of these programs are to reduce environmental degradation, to create green spaces, to supply the enormous demand for forest products, and to conserve biodiversity. Note that the data analyzed here do not distinguish between natural forests and plantations. Rubber, pulp, fruit tree, and eucalyptus plantations have replaced large amounts of natural forest in our study area and plantations start to provide returns. In the context of rapid urbanization, land-cover change is deeply influenced by policy changes and socioeconomic development [23,43]. A previous study examined the tree-cover loss in SWC over the period of 2001–2016, providing an up-to-date report on forest conversion trends in SWC. The finer resolution observation and monitoring of global land cover map indicated that forest and cultivated land were the dominant land cover following forest loss in the loss area; 77.7% of the lost areas were regenerated into forest land and 12% were transformed into cultivated land [50]. Thus, the results of net forest loss will be complex due to the harvesting cycle of timber and selective logging. In a word, the role of market and policy, which transformed land use, improves our understanding of the causes of forest loss at regional/national scales and can be used to broadly inform strategies to improve forest management and enforcement.

### 4.4. Potential Reasons for Deforestation

Identifying the causes of changes in forest dynamics (deforestation and reforestation) is critical to developing measures for the sustainable management of forest resources. The drivers that lead to forest change are multifaceted and combined with different factors. Current research on the driving mechanism of forest change in SWC mainly divides the driving factors into two categories: natural factors and human factors. Wang et al., found that after the implementation of the reform of the collective forest-tenure system, the forest-reduction area in Chongqing, Yunnan, Guangxi, and other southern provinces increased rapidly according to GFC, which explained the impact of policies on forest-resource change and was consistent with our research results [51]. A study on the changes in vegetation net

primary productivity (NPP) between 2001 and 2021 in the southwestern region shows that the regions with a decreasing trend in vegetation NPP are mainly distributed in Yunnan, Guizhou, and Guangxi Province [52]. Agricultural expansion, commercial logging, and ecological projects have also affected the forests of Southwest China to varying degrees. It is a major challenge to preserve forests, biodiversity, and crucial ecosystem services while increasing food production and promoting economic development. A previous study has found that expanding plantations are the main cause of forest loss in SWC [53] and the implementation of ecological engineering further alleviates the degradation of forest resources [54]. Topography, cities, and river systems are also important factors affecting the regional distribution of forest loss. In general, altitude and slope play an important role in vegetation health and growth. The complex terrain of the land surface provides a variety of living conditions for vegetation. Altitude has a certain impact on temperature, humidity, light, and other factors. With the increase in altitude, the temperature gradually decreases, humidity gradually increases, and light intensity gradually increases. The slow slope and steep relief of the terrain mainly affect the loss and accumulation of water and soil and the change of the terrain will have a great impact on the above various climate factors and soil changes, and then affect the growth, development, and distribution of vegetation. Remote and high-altitude areas restrict people's access to the location of the timber. Zeng et al., found that the deforested areas are located mostly in lowlands and flatter areas and with respect to the slope and elevation of areas and forest loss, it is clear that the loss has shifted to much lower and flatter areas [27,28,55]. Another study of SWC revealed altitude and slope are the important terrain factors in the spatial distribution of forest loss, as the altitude and slope increase, the loss of forest area decreased and has high spatial consistency with the distribution of forests [50]. The forest-loss amount is inversely related to the distance from urban residential areas and rivers, as the distance from cities and rivers increases, the deforestation decreased [27]. In addition, the global forest loss caused by forest fires has increased dramatically and frequent fires are considered to be one of the main driving factors affecting forest loss. There is the second largest natural forest area in China and it is also the most serious forest-fire area. Compared with the northern areas, the terrain here is complex, with a high forest-coverage rate per unit area, high personnel density, and various fire sources; it is more prone to fire, which poses a threat to the sustainable development of forest resources [56]. After a fire, the burned land is always covered by herbs first and then replaced by shrubs. The final recovery effect of vegetation is also closely related to the fire intensity, duration, vegetation type, fire season, and other factors. According to the statistics of the National Bureau of Statistics, from 2004 to 2019, a total of 34,011 fires occurred in SWC, covering an area of 467,100 ha. According to the finer resolution observation and monitoring of global land cover map, more than 4% of forest loss was followed by grassland and shrubland over the study period. It is likely that most of this loss corresponds to natural occurrences such as fire and pests [50]. Wu et al., also point out that, in recent years, the increase in fire frequency and fire-site area has destroyed a large amount of forest vegetation [40].

## 5. Conclusions

While a relatively small region, this study in Southwest China provides an important reference point, as the deforestation there has occurred across various regions, and has cast negative impacts on the ecological and socioeconomic systems in the region [8]. GFC datasets have great potential to provide trend data for long-term forest-cover change (loss and gain) with a consistent methodology. Our analysis of the patterns of deforestation in tree cover areas provides essential information to identify efficient methods for managing legal as well as illegal logging. Such quantitative metrics can also provide clues for the potential trajectory and location of future forest changes and may act as guidance to the protection and planning of forest resources for policymakers.

In summary, the results showed that there was a massive loss of forest areas in SWC, especially in the southeastern part. Small-scale (patch size less than 5 ha) deforestation was

widespread and rising across the entire region, despite the mean patch size of forest loss in the region is smaller than 1 ha. High-intensity deforestation had shifted to Sichuan and Chongqing and it is important to enforce inspections of this field since the loss events were more linked to anthropogenic activity, rather than natural disturbances.

In addition, the users should take a conservative view of the data and recognize that an enormous amount of afforestation effort may be around shrubs or narrow strips of trees. However, planting trees is not the same as gaining forests; different tree cover types will typically provide diverse niche spaces and a multitude of ecosystem services (natural forest). It is important to be aware that GFC datasets may consider some selective logging activities as forest loss and have limited applicability in the estimation of deforestation areas. Therefore, causal attribution of specific causes should be interpreted with much caution. In future research, validations and assessments of forest changes need to combine data on the driving forces with very high-resolution imagery or ground-level field data [57], and combine the data that distinguish between primary forests, plantation forests, and commercial forests, and supplement forest-gain data, so as to further refine and precision the forest-change situation and make the analysis more complete.

**Author Contributions:** Conceptualization, X.H. and Y.Z.; methodology, S.W.; formal analysis, S.W. and Y.Z.; investigation, S.W.; data curation, S.W.; writing—original draft preparation, Y.Z. and S.W.; writing—review and editing, X.H.; visualization, S.W.; supervision, X.H.; project administration, X.H.; funding acquisition, X.H. All authors have read and agreed to the published version of the manuscript.

**Funding:** This research was funded by the National Natural Science Foundation of China (NSFC) project, grant number 41771361, 41971079.

**Data Availability Statement:** The GFC dataset used in this study is available at https://glad.earthengine.app/view/global-forest-change (accessed on 10 June 2023).

**Acknowledgments:** We would like to thank the editors and anonymous reviewers for their comments and constructive suggestions.

**Conflicts of Interest:** The authors declare no conflict of interest.

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
