# Peer review of "Spatial–Temporal Dynamics of Forest Extent Change in Southwest China in the Recent 20 Years"

_forests, doi:10.3390/f14071378_

Round 1

Reviewer 1 Report

This is merely a descriptive study of the deforestation extent in the area of interest. There is a lack of data on the causes of this phenomenon. In particular, it would be important to answer the following questions:

Is there a relationship between deforestation rate and altitude?

Are some vegetation/forest types more at risk from deforestation? Is there a possibility to protect such types?

How are deforested areas used? What type of land use/cover occurs on deforested patches?

Describe economic and political frame of forestry management, because deforestation is mainly political process.

Chapter 2.2.1 – Which software was used (especially for GIS)

Chapter 2.2.2 – Patches are not defined

Line 98 – erroneous title

Better paper title would be

Spatial-temporal dynamics of forest extent change in Southwest China in recent 20 years

or

Spatial-temporal dynamics of forested area change in Southwest China in recent 20 years

Author Response

Dear reviewers:

Thank you very much for your comments and professional advice. These opinions help to improve academic rigor of our article. Based on your suggestion and request, we have made corrected modifications on the revised manuscript. We hope that our work can be improved again. Furthermore, we would like to show the details as follows:

Reviewer 1#

Comment 1 Is there a relationship between deforestation rate and altitude?

The author’s answer: In general, altitude and slope play important roles in vegetation health and growth for land plants. The complex terrain on the land surface provides a variety of living conditions for vegetation. Altitude has a certain influence on temperature, humidity, light and other factors. With the increase of altitude, temperature gradually decreases, humidity gradually increases, and light intensity gradually increases. The gentle slope and steep relief of the terrain mainly affect the loss and accumulation of soil and water. The change of topography will have a great impact on the above mentioned climatic factors and soil changes, and then affect the growth, development and distribution of vegetation. The occurrence area of forest change was consistent with the main distribution area of forest. The deforestation mainly occurred in the low altitude and gentle slope area, and the loss gradually shifted to the area with lower altitude and slower slope. With the increase of elevation and slope, the area of forest loss gradually decreases (Chapter 4.4, Line 418-420).

Comment 2 Are some vegetation/forest types more at risk from deforestation? Is there a possibility to protect such types?

The author’s answer: Rubber, pulp, fruit tree and Eucalyptus plantations have replaced large amounts of natural forest in our study area. The state has carried out ecological protection projects such as "returning farmland to forest" and "protecting natural forests", aiming to play the dual role of ensuring economic development and protecting the environment, and effectively promoting forest restoration, more details in Chapter 4.3.

Comment 3 How are deforested areas used? What type of land use/cover occurs on deforested patches?

The author’s answer: our previous study have indicated that forest and cultivated land are mainly land use/cover type in the loss area. GlobeLand30 data was used to detect and analyze land cover change in Southwest China, and to reveal the relationship between deforestation processes and other types of land cover change(Chapter 4.4 Line 391-393).

[50] Wang, S.; Lai, P.; Hao, B.; Ma, M.; Han, X. Remote Sensing Monitoring and Spatio-temporal Pattern of Deforestation in Southwest China from 2001 to 2019. Remote Sensing Technology and Application. 2021, 36(03):552-563.

Comment 4 Describe economic and political frame of forestry management, because deforestation is mainly political process.

The author’s answer:  We have supplemented Chapter 4.3 to address your concerns and hope that it is now clearer, Please see Chapter 4.3 of the revised manuscript.

Comment 5 Chapter 2.2.1 – Which software was used (especially for GIS)

The author’s answer: Google Earth Engine((http://earthengine.google.com/) (Chapter 2.2.1, Line113).

Comment 6 Chapter 2.2.2 – Patches are not defined

The author’s answer: We have defined patches in our study, patches represent areas where forest loss events occur in our analysis (Chapter 2.2.2, Line131).

Comment 7  Line 98 – erroneous title

 The author’s answer: We have fixed the error.

Comment 8 Better paper title would be Spatial-temporal dynamics of forest extent change in Southwest China in recent 20 years or Spatial-temporal dynamics of forested area change in Southwest China in recent 20 years

The author’s answer: Thank you very much. We have changed to Spatial-temporal dynamics of forest extent change in Southwest China in recent 20 years

Reviewer 2#

Comment 1 However, there are two obvious issues with this study: firstly, there is only data on forest loss and no data on forest area increase. For example, in Figure 2a, the data on the increased forest patches and their areas should be discussed together in order to clarify the reasons for forest land loss. Secondly, there was no discussion or comparison of the rationality of the research results. The reasons for forest loss should be analyzed based on the results of forest inventory, relevant research results in the area, as well as local yearbooks and forest resource bulletins. At present, the majority of the author's discussions are speculative and have not been explained in conjunction with actual data or related scientifically findings.

The author’s answer: In response to the first question, we supplemented the forest gain data, and landscape metrics of forest area increase, the forest increase data refers to the total revenue from 2001 to 2012 in GFC product(Chapter 3.1, Line 205-208). In response to the second question, we supplemented chapter 4.4 to analyze the reasons for forest loss and make discussion of the rationality of our research results with others relevant research results in SWC(Chapter 4.4).

Comment 2 Results section: so many discussion contents present there. Just focus on what does the figures or tables present.

The author’s answer: Revised, We have made modifications to the results section and removed unnecessary discussion contents (Chapter 3.2, Line 256-257; Chapter 3.3 Line 273-281).

Comment 3 Line 301: The results agree with the analysis of..., missed some words.

The author’s answer: We 've corrected the typo, the results agree with the analysis of Wu et al.

[40] Wu, Z.; Yan, S.; He, L.; Shan, Y. Spatiotemporal changes in forest loss and its linkage to burned areas in China. J. For. Res. 2019, 10.1007/s11676-019-01062-0.

Comment 4 Line 332-333: This appears to be associated with the mining activities, where it is known as the “Western Mining Captial”.  It is better to have evidences for this speculation, such as using remote sensing interpretation, or from local yearbooks, relevant research papers, etc.

The author’s answer:  We supplemented relevant government documents. On November 12, 2013, the Ministry of Land and Resources issued the Announcement on the Establishment of the Third Batch of Integrated Exploration Zones (No. 18 of 2013), which officially listed the "Chengkou County Manganese Mine Integrated Exploration Zone in Chongqing" as the third batch of integrated exploration zones in China. The Chengkou Party Committee and Government have vigorously developed mining and actively promoted the construction of new industrialization. In 2017, Chongqing Chengkou Industrial Park purchased 80 acres of land and invested 67.8 million yuan to launch an annual production of 100000 tons of barium sulfate project(Chapter 4.2, Line 330-337).

We would like to thank the referee again for taking the time to review our manuscript.

Yours sincerely,

5,Jun,2023

Reviewer 2 Report

Based on the GFC database, the study indicates the forest loss in the Southwest China, as well as the size and hotspots of the lost forest patches, and explores the reasons for the formation of the above problems. These studies provide evidences for understanding the trend and causes of forest loss in the southwestern region of China, and also provide reference for the next step of forest protection and restoration.

However, there are two obvious issues with this study: firstly, there is only data on forest loss and no data on forest area increase. For example, in Figure 2a, the data on the increased forest patches and their areas should be discussed together in order to clarify the reasons for forest land loss. Secondly, there was no discussion or comparison of the rationality of the research results. The reasons for forest loss should be analyzed based on the results of forest inventory, relevant research results in the area, as well as local yearbooks and forest resource bulletins. At present, the majority of the author's discussions are speculative and have not been explained in conjunction with actual data or related scientifically findings.

Results section: so many discussion contents present there. Just focus on what does the figures or tables present.

Line 301: The results agree with the analysis of..., missed some words.

Line 332-333: This appears to be associated with the mining activities, where it is known as the “Western Mining Captial”.  It is better to have evidences for this speculation, such as using remote sensing interpretation, or from local yearbooks, relevant research papers, etc.

English language is good to read.

Author Response

(The authors gave the same response as above.)

Round 2

Reviewer 1 Report

Answers of three questions are not incorporated in the text:

Is there a relationship between deforestation rate and altitude?

Are some vegetation/forest types more at risk from deforestation? Is there a possibility to protect such types?

How are deforested areas used? What type of land use/cover occurs on deforested patches?

Is there any evidence (source) for statistics at lines 386-396 in Discussion?

Author Response

Response to the Review Comments

Dear reviewer:

We are very sorry for our negligence of not making clear modifications according to the reviewer’s suggestions, and we would like to thank you again for your careful reading, helpful comments, which has significantly improved the presentation of our manuscript. We have carefully considered all comments from the reviewers and revised our manuscript accordingly. The manuscript has also been double-checked, and the typos and grammar errors we found have been corrected. In the following section, we summarize our responses to each comment from the reviewers. We believe that our responses have well addressed all concerns from the reviewers.

Reviewer 1#

Comment1: Is there a relationship between deforestation rate and altitude?

Response: Thank you for this valuable feedback. We have added some related research results to explain the relationship between deforestation rate and altitude. Previous studies founded that the deforested areas are located mostly on lowlands and flatter areas(Xiong et alZeng et alWu et al) and with respect to the slope and elevation of areas and forest loss, as the altitude and slope increase, the loss of forest area decreased in our study area(Wang et al).In the revised manuscript, we have made modifications in the discussion section (page 12, section 4.4, line 416-432) to clarify this.

[27]. Xiong, B.; Chen, R.; Xia, Z.; Ye, C.; Anker, Y. Large‐scale deforestation of mountainous areas during the 21st Century in Zhejiang Province. Land Degrad. Dev. 2020, 31, 1761-1774. doi: https://doi.org/10.1002/ldr.3563

[28]. Zeng, Z.; Gower, D.B.; Wood, E.F. Accelerating forest loss in Southeast Asian Massif in the 21st century: A case study in Nan Province, Thailand. Glob Chang Biol 2018, 24, 4682-4695. doi: https://doi.org/10.1111/gcb.14366

[50]. Wang, S.; Lai, P.; Hao, B.; Ma, M.; Han, X. Remote Sensing Monitoring and Spatio-temporal Pattern of Deforestation in Southwest China from 2001 to 2019. Remote Sensing Technology and Application. 2021, 36(03):552-563.

[55]. Yu, X.J. Forest landscape pattern change and driving force analysis in Sichuan Province. Sichuan Agricultural University: Sichuan, China, 2017. (In Chinese)

Comment2: Are some vegetation/forest types more at risk from deforestation? Is there a possibility to protect such types?

Response: We discussed relevant issues in section 4.3 and section 4.4. As we know, deforestation is mainly political process, especially in China, deforestation has an important connection with the policies implemented by the state for many years. An important limitation of our assessment of deforestation is , in our study, Forests include both native forests and plantations, which limits our ability to track mixed agriculture classes and the changes in different types of forests. According to Wang et al, 77.7% of the lost areas were regenerated into forest land, which means plantation forest are more at risk from deforestation. In the revised manuscript, we have made modifications in the discussion section (page 11, section 4.3, line 388-397; page 12, section 4.4, line 440-449) try to answer this question.

Comment3: How are deforested areas used? What type of land use/cover occurs on deforested patches?

Response: Thanks for your question. As far as we know, our study region is responsible for a large part of the China’s ‘real estate’, and exports more wood products than any other regions. There are many plantations in our study area especially in Guangxi and Yunnan Province, logging is a dominant driver of loss. According to Sarathchandra et al , plantations have replaced large amounts of natural forest in our study area. And Wang et al also pointed out that forest and cultivated land were the dominant land cover following forest loss in the loss area, 77.7% of the lost areas were regenerated into forest land, and 12% were transformed into cultivated land. In the revised manuscript, we have supplemented some research results to address your concerns and hope that it is now clearer, please see page 11, section 4.3, line 388-393 .

[50]. Wang, S.; Lai, P.; Hao, B.; Ma, M.; Han, X. Remote Sensing Monitoring and Spatio-temporal Pattern of Deforestation in Southwest China from 2001 to 2019. Remote Sensing Technology and Application. 2021, 36(03):552-563.

[53]. Sarathchandra, C.; Abebe, Y.; Worthy, F.; Wijerathne, I.; Ma, H.; Bi, Y.; Guo, J.; Chen, H.; Yan, Q.; Geng, Y.;Impact of land use and land cover changes on carbon storage in rubber dominated tropical Xishuangbanna, SouthWest China, EcosystemHealth and Sustainability. 2021, 7(1):14. doi:https:// doi/10.1080/20964129.2021.1915183

Comment4: Is there any evidence (source) for statistics at lines 386-396 in Discussion?

Response: In the revised manuscript, we have expanded 3 references published in recent years (page 11, section 4.3, line 388, line 393). The results of these relatively research will improve the comprehensiveness of our manuscript.

[23]. Leblois, A.; Damette, O.; Wolfersberger, J. What has driven deforestation in developing countries since the 2000s? Evidence from new Remote-Sensing data. World Dev. 2017, 92, 82-102. doi: https://doi.org/10.1016/j.worlddev.2016.11.012

[43]. Liu, Y.; Feng, Y.; Zhao, Z.; Zhang, Q.; Su, S. Socioeconomic drivers of forest loss and fragmentation: A comparison between different land use planning schemes and policy implications. Land Use Pol. 2016, 54, 58-68. doi: https://doi.org/10.1016/j.landusepol.2016.01.016

[50]. Wang, S.; Lai, P.; Hao, B.; Ma, M.; Han, X. Remote Sensing Monitoring and Spatio-temporal Pattern of Deforestation in Southwest China from 2001 to 2019. Remote Sensing Technology and Application. 2021, 36(03):552-563.

Thank you very much for your attention and time. Look forward to hearing from you.

Yours sincerely

9, Jun, 2023
